# A predictive approach to enhance time-series forecasting

Skye Gunasekaran[1], Assel Kembay [1], Hugo Ladret[2], Rui-Jie Zhu[1], Laurent Perrinet [3], Omid Kavehei [4] & Jason Eshraghian [1] ✉

Accurate time-series forecasting is crucial in various scientific and industrial domains, yet deep learning models often struggle to capture long-term dependencies and adapt to data distribution shifts over time. We introduce Future-Guided Learning, an approach that enhances time-series event forecasting through a dynamic feedback mechanism inspired by predictive coding. Our method involves two models: a detection model that analyzes future data to identify critical events and a forecasting model that predicts these events based on current data. When discrepancies occur between the forecasting and detection models, a more significant update is applied to the forecasting model, effectively minimizing surprise, allowing the forecasting model to dynamically adjust its parameters. We validate our approach on a variety of tasks, demonstrating a 44.8% increase in AUC-ROC for seizure prediction using EEG data, and a 23.4% reduction in MSE for forecasting in nonlinear dynamical systems (outlier excluded). By incorporating a predictive feedback mechanism, Future-Guided Learning advances how deep learning is applied to time-series forecasting.

In recent years, deep learning models have been increasingly applied to time-series forecasting, leveraging their ability to model complex, nonlinear relationships within data[1]. Despite these advancements, challenges remain in accurately capturing long-term dependencies due to inherent stochasticity and noise in signals. Time-series data involve complex temporal dynamics and often exhibit non-stationary behaviors. Additionally, they are frequently subject to external influences and perturbations that introduce abrupt changes in the data patterns, making long-term forecasting difficult. As a result, even advanced deep learning models face difficulties when tasked with long-term predictions[2,3].

Complementary to these deep learning approaches, classical time-series methods have long used threshold-based adaptation to capture sudden distributional shifts. Early methods such as the Page-Hinkley test and the Drift Detection Method (DDM) formalize this by keeping a running estimate of error statistics, and raising an alarm when a significant change in data distribution is observed[4,5]. Once drift

is detected, models are either fine-tuned on recent labeled examples or retrained from scratch on a sliding window of past data. This threshold-retraining approach has shown practical performance in domains ranging from anomaly detection to predictive maintenance[6], but it can suffer from abrupt resets, loss of long-term knowledge, and sensitivity to hyperparameter choices for the error threshold.

Beyond these classical methods, several self-supervised approaches use future prediction as a pretext task: given an input $x_t$, they learn to reconstruct $x_{t+n}$. This includes applications from video frame prediction[7] to masked audio modeling[8]. However, because they decouple pretraining from online correction, they do not incorporate continuous feedback from each new observation. As a result, their forecast errors cannot be dynamically adjusted as more data arrives.

To address these challenges, we introduce Future-Guided Learning (FGL), an approach that draws on predictive coding and employs a dynamic feedback mechanism to enhance time-series event forecasting. By leveraging a future-oriented forecasting model that guides a

[1]Department of Electrical and Computer Engineering, University of California, Santa Cruz, CA, USA. [2]Friedrich Miescher Institute for Biomedical Research, Basel, Switzerland. [3]Institut de Neurosciences de la Timone, Aix Marseille Univ, CNRS, Marseille, France. [4]School of Biomedical Engineering, The University of Sydney, Sydney, NSW, Australia. ✉e-mail: jsn@ucsc.edu

past-oriented forecasting model, FGL introduces a temporal interplay reminiscent of Knowledge Distillation (KD)[9], where a "teacher" can provide insights that improve a "student" model's long-horizon predictions.

Other works have explored the application of knowledge distillation to sequential data, such as speech recognition[10–12] and language modeling[13], and have excelled at transfer learning and model compression. While these show value in the application of KD to sequential data, it is not used to *enhance* performance over the baseline. KD can be used to enhance how a model handles temporal dynamics and variance in uncertainty across a time horizon.

Importantly, FGL is rooted in the theory of predictive coding, a theory which treats the brain as a temporal inference engine that refines its internal model by minimizing "prediction errors"[14–17]–the discrepancy between expected and actual inputs–over time and across hierarchical layers of abstraction, progressively building internal models of the world[18,19].

Although predictive coding naturally handles spatio-temporal data, it has yet to penetrate mainstream deep learning[20,21]. Neural Predictive Coding frameworks aim to fill this gap by coupling prediction and error-correction in a unified loop. For example, Oord et al.[22] use an autoencoder to forecast future latent representations, and Lotter et al.'s PredNet[23] stacks LSTM cells that propagate and correct layer-wise prediction errors. While these frameworks offer valuable neuroscientific insights, they often emphasize biological plausibility over empirical forecasting performance and tend to be restricted to specific architectures or domains. As a result, it remains challenging to apply them to diverse time-series tasks, thus motivating the need for a more flexible and performance-driven framework, such as FGL.

We evaluate FGL in two settings (see Supplementary Note 4): (1) EEG-based seizure prediction, where FGL boosts AUC-ROC by 44.8% on average across patients; and (2) Mackey-Glass forecasting, achieving a 23.4% MSE reduction. These results show that FGL not only enhances accuracy but also offers a principled way to leverage uncertainty over time, directly aligning with predictive-coding theory.

## Results

We briefly summarize the two domains in which FGL is evaluated:

- *Event prediction*, where a pretrained seizure-detection "teacher" model distills near-future information into a "student" model tasked with early event prediction. We benchmark on two standard EEG datasets (CHBMIT and AES) and report area-under-ROC improvements relative to strong baselines (MViT and CNN-LSTM).
- *Regression forecasting*, in which we reformulate continuous signal forecasting as a categorical task by *discretizing* each true value $x_{t+n}$ into one of $B$ equal-width intervals (or "bins"). The student predicts a distribution over these $B$ bins via softmax-matched to the teacher's softened logits via KL-divergence-while the hard one-hot bin label remains in the cross-entropy term. Final predictions are recovered as the expectation over bin centers, and performance is measured by the resulting mean squared error (MSE). We explore two resolutions ($B = 25$ vs. $B = 50$) to show how bin granularity trades off difficulty against tighter uncertainty bounds.

## Event prediction results

To compare our method with state-of-the-art (SOTA) approaches, we tested FGL against a Multi-Channel Vision Transformer (MViT)[24] and a CNN-LSTM[25,26]. These models are commonly used in medical settings for temporal data and have demonstrated strong efficacy. Further details, including results on false positive rate (FPR) and sensitivity, are provided in Supplementary Note 1.

On the CHBMIT dataset, our results show a significant improvement with FGL compared to the baseline, with an average 44.8% increase in the area under the receiver operating characteristic curve (AUC-ROC), as shown in Fig. 1a–c. Additionally, FGL enhanced predictions across most patients, with the largest gain observed for patient 5 by a factor of 3.84× . The performance dropped only for patient 23 by a factor of 0.80× compared to the CNN-LSTM, but still significantly outperformed the MViT architecture.

On the AES dataset, where the teacher model was trained on a different set of seizure patients and then used on new individuals at test time, FGL still achieved an average performance improvement of 8.9% over the CNN-LSTM, as shown in Fig. 1b. Further details on implementation and preprocessing are provided in Supplementary Note 1.

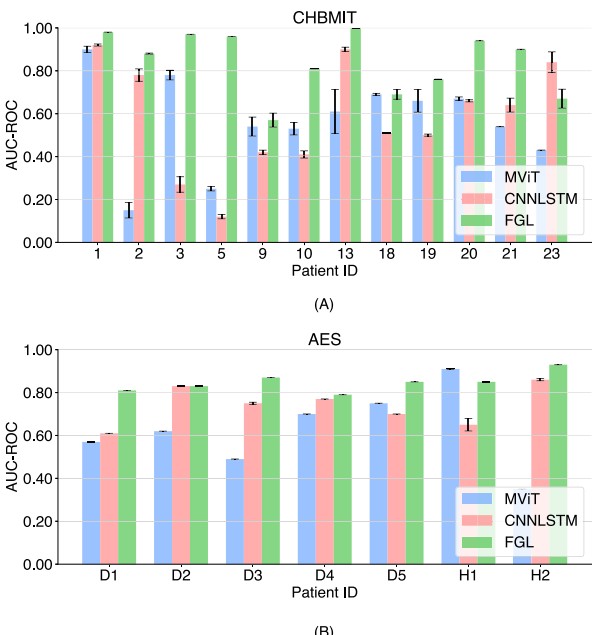

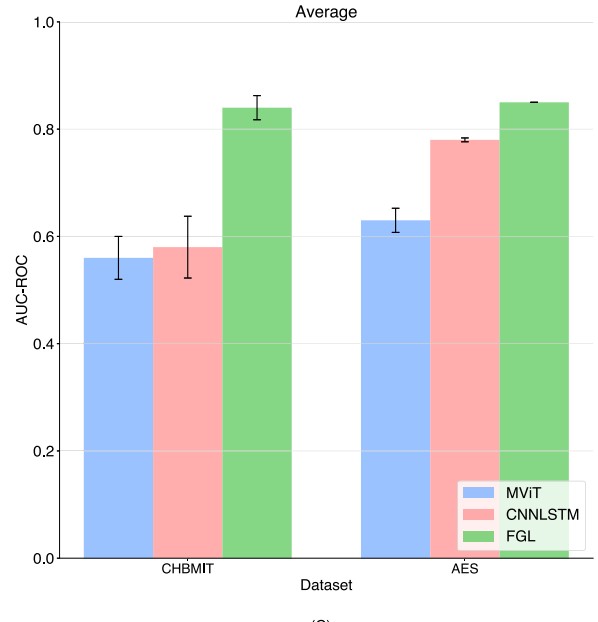

**Fig. 1 | Seizure prediction results. a** CHBMIT, (**b**) AES, and (**c**) dataset averages. Three methods are tested: an MViT, a CNN-LSTM, and FGL. All results were calculated over 3 continuous trials with mean and variance bars displays.

## Regression forecasting results

Across both bin resolutions, incorporating future-guided learning (FGL) yields substantial average MSE reductions versus the baseline. With 25 bins, the baseline's average error is 4.09, while the students trained with FGL achieve 3.64 ($\alpha = 0.0$) and 3.56 ($\alpha = 0.5$)–a 11–13% improvement. When the task is made harder (50 bins), the baseline error jumps to 28.34, but FGL cuts that by more than half to 11.95 ($\alpha = 0.0$) and 11.68 ($\alpha = 0.5$). The consistent edge of the ($\alpha = 0.5$) student over its ($\alpha = 0.0$) counterpart (even if modest) confirms that blending distilled future information with ground-truth targets during training provides a measurable boost.

## Discussion

Overall, our experiments demonstrate that FGL consistently enhances both event-prediction and regression-forecasting tasks by leveraging near-future information to guide long-horizon student models. Below, we examine the distinct gains and underlying mechanisms in each domain, and highlight key trade-offs and robustness benefits.

### Event prediction

On both the CHBMIT and AES datasets, FGL yields substantial increases in AUC-ROC and noticeably tighter error bars compared to our CNN-LSTM and MViT baselines (Fig. 1c). This variance reduction is particularly important in seizure forecasting, where few seizure events per patient can otherwise lead to unstable performance.

A critical factor is the choice of teacher model. With CHBMIT, we trained patient-specific teachers, which captured individual epileptic signatures and delivered the largest average boost (44.8% AUC-ROC) but also higher inter-patient variability. By contrast, our "universal" teachers for AES-pretrained on aggregated UPenn-Mayo seizure data-achieved more modest gains (8.9%) yet produced consistent improvements across all test subjects. Thus, there is a clear trade-off between tailoring guidance to each patient versus exploiting larger, heterogeneous training sets. In practice, one might combine both approaches: use a universal teacher to establish a stable baseline and then fine-tune patient-specific models where data allow.

### Regression forecasting

In the Mackey-Glass experiments, FGL again outperforms the baseline by a wide margin, cutting MSE by 11–13% at 25 bins and by over 50% at 50 bins (Fig. 2). Here, the teacher's short-horizon forecasts serve as a dynamic "upper bound" that guides the student away from catastrophic errors. Rather than penalizing the student harshly whenever the teacher itself errs, our KL-based distillation captures the teacher's uncertainty patterns, yielding a smoother, more informative loss surface.

Examining individual horizons reveals that FGL not only lowers overall error but also produces a gentler, more predictable degradation as the forecast horizon increases. At 25 bins, baseline MSE rises from 1.55 at horizon 2 to 2.30 at horizon 15, whereas FGL students exhibit a flatter slope and smaller peaks. When using 50 bins-where the baseline suffers extreme outliers (e.g., MSE = 195.62 at horizon 13)-both FGL variants cap errors below 12, underscoring dramatic robustness gains in chaotic settings.

### Key insights and future directions

Across both tasks, FGL's effectiveness hinges on two principles: (1) distilling uncertainty from a more confident, near-future teacher and (2) blending distilled signals with ground-truth targets to stabilize learning. Moving forward, we plan to explore hybrid schemes that adaptively weight patient-specific and universal teachers, as well as extensions to multi-horizon and multi-modal forecasting. Finally, integrating FGL with online drift-adaptation methods (e.g., Page-Hinkley) could further enhance resilience to non-stationary environments without discarding long-term knowledge.

## Methods

Traditional KD involves transferring probabilistic class information between two models that share the same representation space. In our approach, we reformulate this student-teacher dynamic by placing the teacher model in the relative future of the student model, introducing a temporal difference in the representation space between them.

Our distillation method follows that of Hinton et al.[9], where the student model is trained using a combination of the cross-entropy loss with the ground truth labels and the Kullback–Leibler (KL) divergence between the softmax outputs of the student and teacher models. This dual objective allows the student to learn from both the true data and the future-oriented predictions of the teacher.

**Notation:** Let $x_t \in \mathcal{X}$ be the input observed up to time $t$, and $y_{t+\ell} \in \mathcal{Y}$ the target at horizon $\ell$. We denote by

$$T_\phi : \mathcal{X} \to \mathbb{R}^C \quad \text{and} \quad S_\theta : \mathcal{X} \to \mathbb{R}^C$$

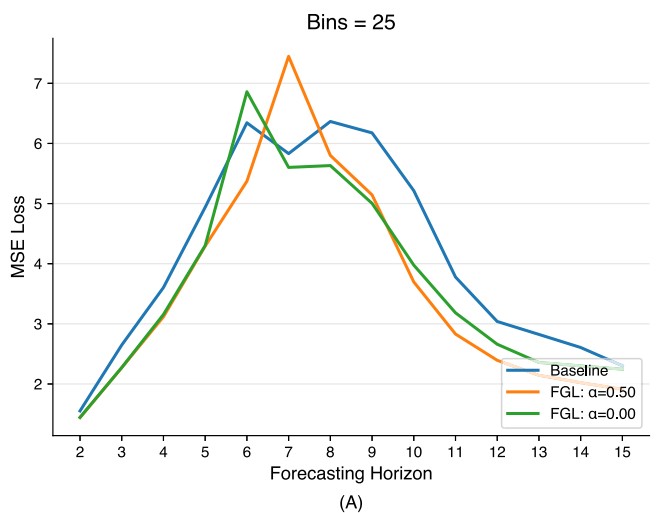

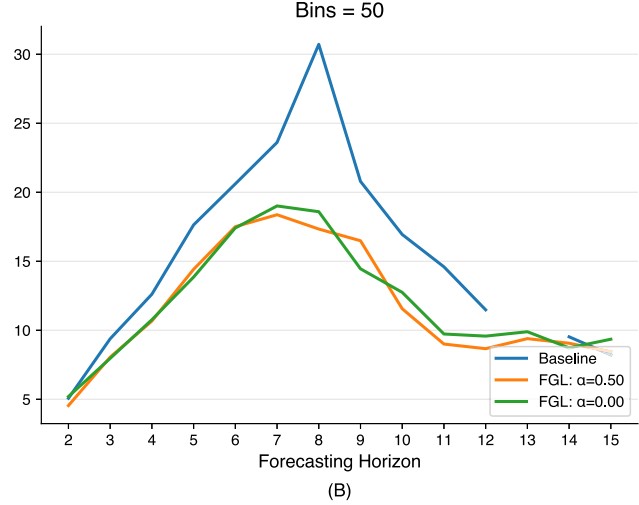

**Fig. 2 | Mackey-Glass forecasting results. A** 25 bins and (**B**) 50 bins. Results show the MSE loss at each horizon. In (**B**), the baseline MSE at horizon 13 was a large outlier and has been omitted for clarity. **A, B** $\alpha = 0.00$ indicates the student trained using only the distilled label, and $\alpha = 50$ indicates a balance of distilled and ground-truth labels. A table of results is available in Supplementary Note 2.

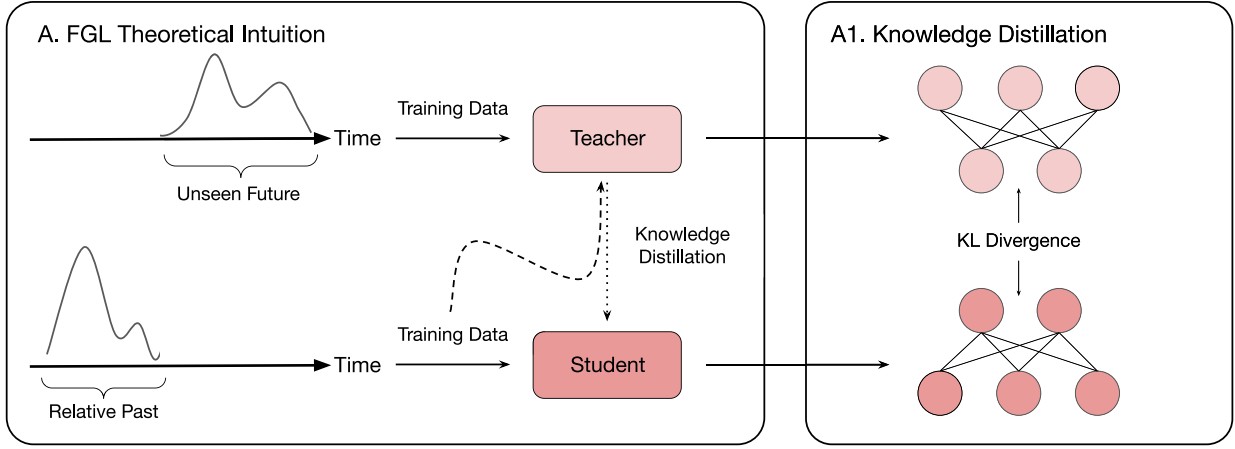

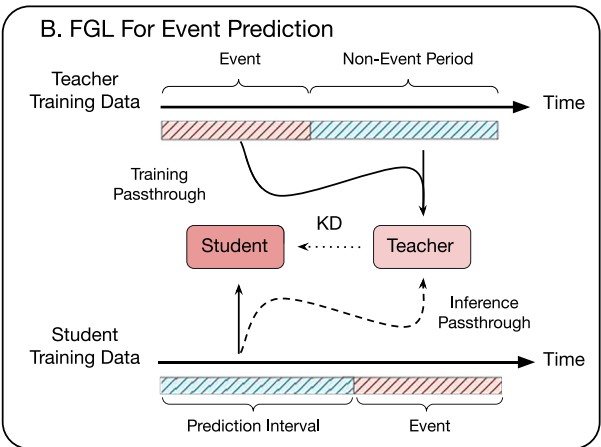

**Fig. 3 | Overview of FGL and its applications. A** In the FGL framework, a teacher model operates in the relative future of a student model that focuses on long-term forecasting. After training the teacher on its future-oriented task, both models perform inference during the student's training phase. The probability distributions from the teacher and student are extracted, and a loss is computed based on Eq. (1). **A1** Knowledge distillation transfers information via the Kullback–Leibler (KL) divergence between class distributions. **B** In an event prediction setting, the teacher is trained directly on the events themselves, while the student is trained to forecast these events. Future labels are distilled from the teacher to the student, guiding the student to align more closely with the teacher model's predictions, despite using data from the relative past. **C** In a regression forecasting scenario, the teacher and student perform short-term and long-term forecasting, respectively. Similar to event prediction, the student gains insights from the teacher during training, enhancing its ability to predict further into the future.

two neural network models parameterized by $\phi$ and $\theta$, producing $C$-dimensional *logits*. The teacher $T_\phi$ forecasts $n$ steps ($y_{t+n}$), while the student $S_\theta$ forecasts $n+k$ steps ($y_{t+n+k}$).

**Claim 1.** (Future-Guided Learning (FGL)) Let

$$T_\phi(x_t) \approx y_{t+n}, \quad S_\theta(x_t) \approx y_{t+n+k},$$

be the logits of a teacher forecasting $n$ steps ahead and a student forecasting $n+k$ steps. FGL trains the student by minimizing:

$$\mathcal{L}_{FGL}(\theta) = \underbrace{\alpha \mathcal{L}_{task}\big(S_\theta(x_t), y_{t+n+k}\big)}_{\text{task loss}}$$
$$+ \underbrace{(1-\alpha)\tau^2 KL\left(\sigma\left(\frac{T_\phi(x_{t+k})}{\tau}\right) \,\|\, \sigma\left(\frac{S_\theta(x_t)}{\tau}\right)\right)}_{\text{future-guided distillation}}, \quad (1)$$

where $\sigma$ is softmax, $\tau$ the distillation temperature, and $0 < \alpha < 1$ balances ground truth and teacher guidance. By aligning the teacher's $n$-step logits at $t+k$ with the student's ($n+k$)-step logits at $t$, FGL transfers near-future uncertainty to the long-horizon forecaster.

The first term in $\mathcal{L}_{FGL}$ ensures the student learns to match true labels at $t+n+k$. The second term softly aligns the student's long-horizon distribution with the teacher's nearer-horizon distribution, effectively distilling "future" uncertainty Fig. 3a.

In practice, we set $\mathcal{L}_{task}$ to cross-entropy (for classification) or MSE (for regression). We pretrain $T_\phi$ on its $n$-step task, then freeze it while training $S_\theta$ under the combined FGL loss. Figure 3b illustrates the offset in the data flow.

In classic distillation[9], both teacher and student forecast the *same* horizon from identical inputs. FGL instead introduces a *temporal offset*: the teacher's logits come from a shifted time step $t+k$, providing an extra supervisory signal drawn from the near future.

### CHBMIT: patient-specific FGL
On the CHBMIT dataset, FGL was performed by first pre-training a unique teacher model for seizure detection on each patient for 50 epochs. During this pre-training phase, preictal segments were excluded, as their inclusion would reduce the uncertainty we aim to distill during the student training phase. Next, during the training of the student model over 25 epochs, each data point was fed into both the student and teacher models, and the corresponding class probabilities were obtained. The loss function defined in Eq. (1) was then used to compute the student model's loss. A range of $\alpha$ values was tested to

balance the contributions of the cross-entropy and KL divergence components of the loss function. The optimal $\alpha$ for each patient was selected based on a hyperparameter sweep, with detailed results presented in Supplementary Fig. 1. Different values of $\alpha$ were tested to balance the cross-entropy and KL divergence components of the loss, with the optimal value selected through a hyperparameter sweep for each patient. The temperature parameter for the KD loss was fixed at $T = 4$. We used the SGD optimizer with a learning rate of $5 \times 10^{-4}$. Both the student and teacher models were implemented as CNN-LSTM architectures.

## AES: patient non-specific FGL

When future data from the same distribution as the student model is unavailable, it is still possible to train a teacher model using future data from an out-of-distribution source. We test this approach on the AES dataset, which contains only preictal samples without any labeled seizures. Due to the lack of corresponding seizure data, we use a separate dataset: the UPenn and Mayo Clinic seizure dataset[27] to create the teacher model.

The AES dataset comprises recordings from 5 dogs and 2 humans, while the UPenn and Mayo Clinic datasets include 4 dogs and 8 human patients. Given this discrepancy, we constructed two "universal teachers" from the latter dataset: one based on dog seizures and the other on human seizures. These universal teacher models were trained using a combined set of interictal data from selected patients, interspersed with randomly sampled seizures from a diverse pool of patients. This approach allows the teacher model to learn generalized seizure features from a wide variety of cases.

A challenge in using different data sources for the teacher model was the inconsistency in data characteristics and formats between datasets. We addressed this issue by selecting the top $k$ most significant EEG channels for each teacher model based on their contribution to seizure detection scores[28]. The number of selected channels was then adjusted based on the requirements of each student model, ensuring compatibility and effective knowledge transfer. As with the CHBMIT dataset, an ablation study of the influence of $\alpha$ is provided in Supplementary Figs. 1 and 2.

## Regression forecasting implementation

To quantify uncertainty in our regression forecasts, we map each continuous target value $x_{t+n}$ onto a discrete probability distribution over $B$ equally-spaced bins (Fig. 4a). Concretely, we first partition the full range of observed $x$-values into $B$ contiguous intervals, then represent the "true" target by a one-hot vector indicating the bin that contains $x_{t+n}$. Our network's final layer has $B$ neurons with softmax activations, so its output $\mathbf{p} \in \Delta^{B-1}$ encodes a categorical distribution over these intervals.

During distillation, the teacher supplies a softened $\mathbf{p}^{(T)}$, which the student matches via a KL-divergence loss; the hard one-hot label is still used in the cross-entropy term. Increasing $B$ narrows each interval—leading to finer-grained value ranges and tighter uncertainty bounds—at the expense of making the classification task more difficult. This binning strategy thus lets us reduce a regression problem to a probability-distribution prediction, enabling us to leverage standard classification losses (cross-entropy + KL) while still recovering a real-valued forecast (e.g., by taking the expectation over bin centers).

In addition, we reformulate the tasks of the teacher and student models to better align with regression objectives. The teacher model performs next-step forecasting, while the student model focuses on longer-term predictions. This difference in forecasting horizons explicitly enforces a variance in timescales between the models. More specifically, given an input sample at $x_t$, the student model aims to predict $n$ steps into the future, targeting $x_{t+n}$. In contrast, the teacher model is tasked with predicting the immediate next step and is provided with the input data at $x_{t+n-1}$.

Similar to event prediction, the teacher model is pretrained first, followed by inference within the student training loop, where the loss defined in Eq. (1) is computed. During model testing, we select the neuron with the highest probability and compute the corresponding MSE, aligning this approach with traditional regression evaluation methods. Full experimental details can be found in Supplementary Note 2.

## Future guided learning and predictive coding

The teacher and student models can be described more precisely using a Bayesian prediction framework. The teacher, with access to $n$ future points, has predictive density

$$p_T(x_{t+n}) = \int f(x_{t+n} \mid \theta)\, \pi(\theta \mid x_{1:t+n-1})\, d\theta,$$

while the student, limited to the current window, uses

$$p_S(x_{t+n}) = \int f(x_{t+n} \mid \theta)\, \pi(\theta \mid x_{1:t})\, d\theta.$$

Their divergence is

$$D_{\mathrm{KL}}(p_T \parallel p_S) = \int p_T(x)\, \ln \frac{p_T(x)}{p_S(x)}\, dx,$$

quantifying how much "extra" information the teacher holds over the student as the horizon grows.

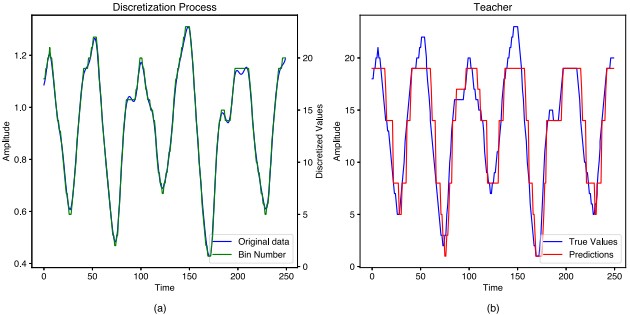
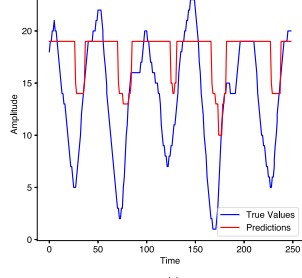
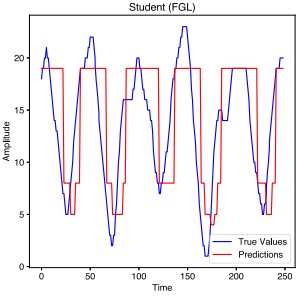

**Fig. 4 | Discretization enables knowledge-distillation on Mackey-Glass. a** After generating the chaotic trajectory, every scalar target $X_{t+\ell}$ is *quantized* into one of $C$ equal-width bins. The regression problem is thus recast as $C$-way classification, so the teacher and student can exchange *soft logits* of identical dimensionality-an essential requirement for the KL-distillation term in FGL. **b** Teacher performs next-step prediction. **c** Baseline performs a 5-step forecast without future guidance. **d** FGL-trained student forecasting the same horizon. By aligning its logit distribution with the teacher's near-future logits, the FGL student captures neighborhood information in bin space, yielding a visibly lower MSE and a smoother reconstruction.

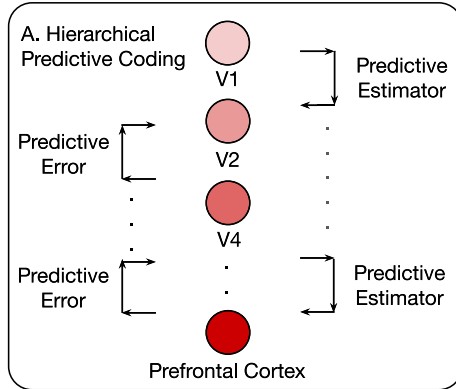

**Fig. 5 | Predictive Coding and FGL. A** Illustration of predictive coding in the brain. Information is received by the primary visual cortex (V1), which then propagates to different areas of the brain with more complex levels of abstraction. This propagation takes the form as a predictive estimator: higher level areas aid in prediction of lower level areas. The difference between the prediction and true output is the predictive error. Areas with lighter colors represent low-level abstractions, where darker colors represent the increasing representational demand (thoughts, movement, etc). **B** Hierarchical FGL propagates information via uncertainty; in other words, FGL substitutes the predictive estimator with the uncertainty of each layer. This uncertainty is conveyed via the KL divergence of between each layer's probability distribution with the successive layer. The difference between each layer's prediction and true output is the predictive error. In both models, areas lower in the hierarchy process information in a delayed manner, as they are the last to receive it. As the demand for complexity increases, so does the predictive error.

Rather than distilling directly from the longest-horizon teacher, we chain predictions through intermediate models at steps $t+1, \ldots, t+n-1$. Each acts as both student (to its predecessor) and teacher (to its successor), propagating uncertainty down the hierarchy.

This approach bears similarity to a potential implementation of hierarchical predictive coding in the brain. Low-level cortical areas, such as the primary visual cortex (V1), function analogously to the intermediate models in hierarchical FGL, as they process detailed sensory inputs over short timescales (e.g., $f(x_{n+t}|x_{n+t-1}, \ldots, x_t)$). In contrast, higher-level cortical areas like the prefrontal cortex correspond to the bottommost student model, $f(x_{n+t}|x_n, \ldots, x_t)$, as they integrate abstract patterns and process temporally delayed information over longer timescales. A visual representation of this hierarchical structure is provided in Fig. 5.

In practice, true posteriors are intractable. We therefore treat each $p_T$ as a variational surrogate $q(v)$ and each $p_S$ as a prior $p(v)$, optimizing the usual ELBO:

$$\ln p(u) = -F + D_{KL}(q(v) \parallel p(v|u)),$$

where $F$ is the free energy (a lower bound on surprise)[20,29]. Finally, assuming Gaussian predictive distributions gives the familiar precision-weighted error form:

$$\ln p(u) \approx -\frac{1}{2}\left[\ln \Sigma_S + \frac{(v_S - \Phi)^2}{\Sigma_S} + \ln \Sigma_T + \frac{(u - g(\Phi))^2}{\Sigma_T}\right].$$

Minimizing this drives the student toward the teacher's richer, future-informed predictions.

## Data availability
All datasets used in this study are publicly available: CHB–MIT Scalp EEG Database: Available at https://physionet.org/content/chbmit/1.0.0/. Kaggle Seizure Prediction Challenge: Available at https://www.kaggle.com/competitions/seizure-prediction. Kaggle UPenn & Mayo Clinic Seizure Detection Challenge: Available at https://www.kaggle.com/competitions/seizure-detection. Synthetic Mackey–Glass time series generated for this work using the standard Mackey–Glass delay differential equation (see Supplementary Note 2). Parameter settings and generation scripts are provided in the accompanying code repository (see Code availability, below); preprocessed train/test splits used in our experiments are included there for reproducibility.

## Code availability
Our code is publicly available at the following github repository: https://github.com/SkyeGunasekaran/FutureGuidedLearning.

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

## Acknowledgements

This work is supported by the US National Science Foundation ECCS:2332166 (J.E.).

## Author contributions

S.G. and J.E. conceived the project, designed the experiments, and drafted the manuscript; S.G. performed all simulations; A.K., H.L., and R.Z. refined the experimental design; H.L. and L.P. derived the predictive-coding equations; O.K. supervised the analysis; J.E. oversaw the study. All authors reviewed the manuscript and approved the final version for submission.

## Competing interests

The authors declare no competing interests.
