## [Transparent Peer Review file · Nature Communications]

A Predictive Approach to Enhance Time-Series Forecasting

Corresponding Author: Professor Jason Eshraghian

Version 0:

Reviewer comments:

Reviewer #1

(Remarks to the Author)

This paper proposes the Future-Guided Learning approach for helping deep learning-based time series event forecasting models capture long-term dependencies and adapt to data distribution drifts over time. It uses a dynamic feedback mechanism involving a detection model that analyzes future data to identify events (teacher) and a forecasting model that predicts these events based on current data (student). When discrepancies occur between the forecasting and detection models, an update is applied to the forecasting model. It is inspired by predictive coding.

Strong points:

- The approach and use-cases are very interesting and relevant to the field
- The paper presents extensive experiments
- Good theoretical background
- The methodology is sound
- Results are reproducible and data is provided

Weak points:

- The paper is difficult to read because the information is compartmentalized in several sections and appendices, and in an order that is not ideal for comprehension
- The paper lacks related works for contextualization
- Experiments are not clear and additional evidence is needed

Main comments:

- At first, the description of the approach seems to defend the use of the "future" to predict the future. But after further reading it is clear that it is using the most recent values to validate the last predictions and adapting the forecasting model if needed.
- What is described as a feedback loop, however, does not seem so different from the idea of monitoring forecasting/detection errors and updating the model when they surpass a threshold. In fact, this process is widely adopted in applications prone to concept drift. There should be an experimental comparison with this basic approach for increasing the validity of the results.
- Also, I am concerned about overfitting. Since the approach may present a high level of adaptivity, it seems prone to overfitting. What were the precautions for avoiding it?

Detailed comments:

- Figure 1 is not cited in the text. Only 1b in the seventh page.
- Section 2 goes to the results before explaining the method. It is very confusing.
- The order of the plots of Figure 4 is wrong.
- The competing methods (MViT and CNN-LSTM) should be minimally introduced. Also the "baseline" method was never defined.
- The paper lacks discussion on related works. It is paramount to understand the approach and its contributions in the context of the current literature.
- It is confusing to have the method and experiment explained after the results. By the time we read the results, we do not know how the experiment was set, nor any details of the data.
- "Continuous values in the regression task are discretized into bins, such that logits can be distilled from teacher to student." The discretization demands better explanation. I do not understand the need or the process adopted to do it. It was not explained in the text.

(Remarks on code availability)

Reviewer #2

(Remarks to the Author)

This paper proposes a future-oriented time series prediction method called FGL(Future-Guided Learning). The core idea is to pre-train a "future-oriented" detection model as a teacher model and then use time series training data to jointly train a student model to achieve long-horizon prediction. Two types of examples are used for validation:

1. Event prediction task – Specifically, predicting seizures based on EEG data.
2. Regression forecasting task – Specifically, forecasting time series data generated by the Mackey-Glass (MG) system.

The FGL method is implemented through the design of the loss function in Eq. (1), which consists of two parts:

- The first part is the cross-entropy loss between the student's output and the true label Y_{true} , which measures the student's ability to learn from true data.
- The second part is the KL divergence between the student's output and the teacher's output (aligned in time through an offset), which measures the student's ability to learn from the teacher model.

This idea is innovative, but the paper fails to provide clear explanations on many key issues, making it difficult to understand.

Major Concerns

The underlying principle is unclear.

Knowledge distillation is a machine learning technique designed to transfer the knowledge from a large, pre-trained "teacher model" to a smaller "student model." A key assumption is that the teacher model should be better, or more powerful than the student model.

However, in this paper, both the teacher and student models are trained on the same set of training data. This means that the teacher's output—whether it's a one-step prediction from $t+n-1$ to $t+n$, or an event detection based on $t+n$ —should not be better than the true label (since Eq. (1) uses Y_{true}) as the ground truth for $t+n$.

So why would the teacher provide additional capabilities beyond the ground truth in the Loss function(1)?

Particularly, in the MG system example, both the teacher model and the student model are trained on the same set of training data. In this example, the teacher model is an one-step predictor while the student model is a 5-step predictor. During the student model's training, the ground truth label at $t+5$ is used in the first term of the loss function, while the second term incorporates the label for $t+5$ generated by the teacher model based on the true data at $t+4$. Notably, under this setup, there is no true "future guidance"—only labels produced by the one-step predictor. Surprisingly, it is reported that when $\alpha=0$ (i.e., only the teacher's output is used in training, with ground truth labels completely excluded), the model achieves the best performance. This is highly counterintuitive because, in the MG system example, the teacher is trained on the same data for one-step prediction. Its prediction accuracy should not surpass the ground truth. Why, then, does relying solely on the teacher's output yield better results than using both the ground truth and the teacher?

Minor Concerns

Workflow is not unclear shown.

- What are the inputs and outputs?
- What are the training and test sets?
- Is the "future" data selected from within the training set?
- How is input/output handled during actual prediction?

(Remarks on code availability)

There are some minor errors in the codes but the results should not be affected.

Version 1:

Reviewer comments:

Reviewer #1

(Remarks to the Author)

The authors understood and well addressed my concerns. I am satisfied with the revised version of the paper.

(Remarks on code availability)

Reviewer #2

(Remarks to the Author)

All my concerns have been answered. The replies to my major comment 2 are encouraged to be integrated into the main text with details.

(Remarks on code availability)

Response to the Editor and Reviewers

Manuscript ID: NCOMMS-25-12507

Title: *Future-Guided Learning: A Predictive Approach to Enhance Time-Series Forecasting*

We would like to thank the Handling Editor for inviting a major revision and for guiding our manuscript through such a rigorous review process. We are grateful to both reviewers for their thoughtful, constructive feedback and are pleased that they both see value in the “future-guided” learning (FGL) technique. We have put significant effort into improving the original presentation of both the core idea and experimental framework, as both reviewers highlighted that these should be expressed more clearly. Reviewer 1 made an important comment regarding how our proposed technique compares against established drift-adaptation and distillation baselines. To address these we have:

- Reorganized the manuscript for a more logical flow—Methods now precede Results, and key concepts are introduced earlier.
- Expanded experimental validation on the Mackey–Glass task, including a supplemental experiment using Page–Hinkley drift retraining.
- Revised the Methodology section to now open with a formal **Claim** that states the FGL loss and training procedure in a single equation. This is followed by an explanation on how the loss can specialize to both classification (cross-entropy) and regression (MSE).
- Provided full experimental details—dataset splits, hyperparameters, metrics tables—and updated publicly available codebase.

All changes are shown in blue in the marked-up manuscript. Below we address each reviewer comment point-by-point.

Response to Reviewer 1

This paper proposes the Future-Guided Learning approach for helping deep learning-based time series event forecasting models capture long-term dependencies and adapt to data distribution drifts over time. It uses a dynamic feedback mechanism involving a detection model that analyzes future data

to identify events (teacher) and a forecasting model that predicts these events based on current data (student). When discrepancies occur between the forecasting and detection models, an update is applied to the forecasting model. It is inspired by predictive coding.

Strong points:

- *The approach and use-cases are very interesting and relevant to the field*
- *The paper presents extensive experiments*
- *Good theoretical background*
- *The methodology is sound*
- *Results are reproducible and data is provided*

Response. We would like to thank the reviewer for both their strong positive feedback as well as their valuable and well-justified criticisms, especially with respect to a) improving the organization and clarity of our manuscript, and b) the need to provide additional baselines. In summary, we have revamped the manuscript based on the Reviewer’s comments and have provided our point-by-point response and summary of modifications below.

Comment 1

“The paper is difficult to read because the information is compartmentalized in several sections and appendices, and in an order that is not ideal for comprehension.”

Response. We appreciate the reviewer’s emphasis on clarity and completely agree. In organizing our first draft, we followed the section order most frequently used in recent *Nature Communications* articles—Results first, then Methods. While that template matches the journal’s default layout, we recognize that, for a methodological contribution such as FGL, interleaving scattered details across the main text and appendix made the story hard to reconstruct and interpret. Accordingly, we have *reorganized* the manuscript as follows:

- Introduction
- Related Work

- Methodology
- FGL For Event Prediction
- FGL For Regression Forecasting
- FGL and Predictive Coding
- Conclusion

By placing all methodological details (formerly spread across appendices) directly before their corresponding results, and by renaming section titles to be more descriptive, we believe the narrative now flows logically and is far easier to follow.

In addition, the *Methodology* section itself has been substantially expanded and formalized. It now begins with a brief, plain-language road map, proceeds to **Claim 1**—which presents the full FGL loss in Equation (1)—and then shows how that same formulation specializes to classification (cross-entropy) and regression (MSE). This rewrite makes the definition of FGL self-contained, easier to cite throughout the paper, and clearer for readers who wish to re-implement the method. The following has been included in **Section 3**:

Notation: Let $x_t \in \mathcal{X}$ be the input observed up to time t , and $y_{t+\ell} \in \mathcal{Y}$ the target at horizon ℓ . We denote by

$$T_\phi : \mathcal{X} \rightarrow \mathbb{R}^C \quad \text{and} \quad S_\theta : \mathcal{X} \rightarrow \mathbb{R}^C$$

two neural network models parameterized by ϕ and θ , producing C -dimensional *logits*. The teacher T_ϕ forecasts n steps (y_{t+n}), while the student S_θ forecasts $n+k$ steps (y_{t+n+k}).

Claim (Future-Guided Learning (FGL)). *Let*

$$T_\phi(x_t) \approx y_{t+n}, \quad S_\theta(x_t) \approx y_{t+n+k},$$

be the logits of a teacher forecasting n steps ahead and a student forecasting $n+k$ steps. FGL trains the student by minimizing:

$$\mathcal{L}_{\text{FGL}}(\theta) = \underbrace{\alpha \mathcal{L}_{\text{task}}(S_\theta(x_t), y_{t+n+k})}_{\text{task loss}} + \underbrace{(1 - \alpha) \tau^2 \text{KL}\left(\sigma\left(\frac{T_\phi(x_{t+k})}{\tau}\right) \parallel \sigma\left(\frac{S_\theta(x_t)}{\tau}\right)\right)}_{\text{future-guided distillation}}, \quad (1)$$

where σ is softmax, τ the distillation temperature, and $0 < \alpha < 1$ balances ground truth and teacher guidance. By aligning the teacher’s n -step logits at $t + k$ with the student’s $(n + k)$ -step logits at t , FGL transfers near-future uncertainty to the long-horizon forecaster.

The first term in \mathcal{L}_{FGL} ensures the student learns to match true labels at $t + n + k$. The second term softly aligns the student’s long-horizon distribution with the teacher’s nearer-horizon distribution, effectively distilling “future” uncertainty Figure 1(a).

In practice, we set $\mathcal{L}_{\text{task}}$ to cross-entropy (for classification) or MSE (for regression). We pretrain T_ϕ on its n -step task, then freeze it while training S_θ under the combined FGL loss. Figure 1(c) illustrates the offset in the data flow.

In classic distillation¹, both teacher and student forecast the *same* horizon from identical inputs. FGL instead introduces a *temporal offset*: the teacher’s logits come from a shifted time step $t + k$, providing an extra supervisory signal drawn from the near future.

Furthermore, **Figure 1(c) has been modified to reflect the new variables t , n , and k :**

Comment 2

“The paper lacks related works for contextualization.”

Response. We agree that stronger literature grounding is essential. We have added a dedicated Related Work section (Sec. 2) that: 1) reviews classical concept-drift and threshold-based retraining strategies, 2) summarizes prior “future-aided” teacher–student schemes in forecasting and self-supervised learning, and 3) highlights recent predictive-coding–inspired deep models. This gives readers a clear map of existing approaches and positions FGL’s continuous, future-teacher feedback as a novel contribution. The following has been included in **Section 2:**

Other works have explored the application of knowledge distillation to sequential data, such as speech recognition^{2–4} and language modeling⁵, and have excelled at transfer learning and model compression. While these show value in the application of KD to sequential data, it is not used to *enhance* performance over the baseline. KD can be used to enhance how a model handles temporal dynamics and variance in uncertainty across a time-horizon.

Figure 1: (A) In the FGL framework, a teacher model operates in the relative future of a student model that focuses on long-term forecasting. After training the teacher on its future-oriented task, both models perform inference during the student’s training phase. The probability distributions from the teacher and student are extracted, and a loss is computed based on Equation 1. (A1) Knowledge distillation transfers information via the Kullback-Leibler (KL) divergence between class distributions. (B) In an event prediction setting, the teacher is trained directly on the events themselves, while the student is trained to forecast these events. Future labels are distilled from the teacher to the student, guiding the student to align more closely with the teacher model’s predictions, despite using data from the relative past. (C) In a regression forecasting scenario, the teacher and student perform short-term and long-term forecasting, respectively. Similar to event prediction, the student gains insights from the teacher during training, enhancing its ability to predict further into the future.

Beyond KD, several self-supervised methods use future prediction as a pretext task: given an input x_t , they learn to reconstruct x_{t+n} . This includes applications from video frame prediction⁶ to masked audio modeling⁷. However, because they decouple pretraining from online correction, they do not incorporate continuous feedback from each new observation. As a result, their forecast errors cannot be dynamically adjusted as more data arrives.

Neural Predictive Coding frameworks aim to fill this gap by coupling prediction and error-correction in a unified loop. For example, Oord et al.⁸ use an autoencoder to forecast future latent representations, and Lotter et al.’s PredNet⁹ stacks LSTM cells that propagate and correct layer-wise prediction errors. While these frameworks offer valuable neuroscientific insights, they often emphasize biological plausibility over empirical forecasting performance and tend to be restricted to specific architectures or domains. As a result, it remains challenging to apply them to diverse time-series tasks, thus motivating the need for a more flexible and performance-driven framework, such as FGL.

Complementary to these continuous feedback loops, classical time-series methods have long used threshold-based adaptation to capture sudden distributional shifts. Early methods such as the Page–Hinkley test and the Drift Detection Method (DDM) formalize this by keeping a running estimate of error statistics, and raising an alarm when a significant change in data distribution is observed^{10,11}. Once drift is detected, models are either fine-tuned on recent labeled examples or re-trained from scratch on a sliding window of past data. This threshold-retraining approach has shown practical performance in domains ranging from anomaly detection to predictive maintenance¹², but it can suffer from abrupt resets, loss of long-term knowledge, and sensitivity to hyperparameter choices for the error threshold. This happens because introducing a separate drift-detection module can generate false alarms—and then fine-tuning exclusively on a narrow sliding window of recent data discards previously learned representations, leading to catastrophic forgetting and high sensitivity to the chosen error threshold. By contrast, FGL integrates feedback from a “future” teacher at training time, bypassing the coarse-grained drift alarm and enabling smoother adaptation.

Comment 3

“Experiments are not clear and additional evidence is needed.”

Response. Regarding the seizure-prediction experiments, we have expanded their details in Appendix A.3: exact dataset splits, hyperparameter settings, and training strategy are discussed. The following has been included in **Appendix A.3:**

Dataset splits. For each patient we gather *all* annotated EEG segments:

1. every inter-ictal (I) segment, and
2. every ictal *or* pre-ictal (S) segment.

We then build a **balanced, chronological stream**

$$I_1 S_1 I_2 S_2 \dots$$

by interleaving the two classes so that each seizure block S_j is immediately preceded by the amount of inter-ictal data required to keep the cumulative class counts equal.¹ This balanced stream is split *contiguously*: the first 65 % forms the training set and the final 35 % the held-out test set. No window, channel, or seizure appears in both splits.

Models compared:

- **Baseline (CNN-LSTM).** batch-norm \rightarrow Conv3D($1 \rightarrow 16, k = (T/2, 5, 5), s = (1, 2, 2)$) \rightarrow max-pool \rightarrow batch-norm \rightarrow Conv3D($16 \rightarrow 32, k = 3, s = 1$) \rightarrow avg-pool($3, 2$) \rightarrow 3-layer LSTM (512 hidden) \rightarrow FC 512 to FC 256 to 2.
- **FGL (ours).** Teacher: Seizure-detector (CNN-LSTM Backbone) Student: Seizure-predictor (CNN-LSTM Backbone) Trained with the Future-Guided loss (Eq. 1); $\alpha \in \{0, 0.1, \dots, 1\}$.
- **MViT.** Multi-scale Vision Transformer per EEG channel: patch size (5, 10), embed dim 128, 4 heads, hidden dim 256, 4 encoder layers, dropout 0.1. Channel embeddings are concatenated and fed to a linear head ($128 \times C \rightarrow 2$).

Optimisation and over-fitting control: All models are trained with Adam ($\text{lr} = 5 \times 10^{-4}$, $\beta_1 = 0.9$, $\beta_2 = 0.999$, $\varepsilon = 10^{-8}$); batch size 32; maximum 25 epochs. To guard against overfitting,

¹The first block is always inter-ictal because patients typically enter the hospital in a non-seizure state.

each experiment is repeated with multiple random initializations on the same dataset, and stability is assessed across runs.

Evaluation: For each patient we conduct three independent trials and report mean \pm std. of sensitivity, false-positive rate (FPR), and AUC-ROC on the test split. The decision threshold is chosen per patient via Youden’s J statistic. Results appear in Table 1; best numbers per patient are shown in **bold**.

Hardware: All models were trained using an NVIDIA RTX 4080 GPU.

Additional performance metrics including false-positive rate, sensitivity, and AUROC—for each model, are reported in Table 1 and Table 2 in the Appendix. These additions ensure full transparency and allow readers to reproduce and interpret our results more confidently.

Furthermore, regarding the Mackey-Glass experiments, we have changed the following:

- Fully trained both baseline and student models for 50 epochs: Previously we ran only 10–20 epochs; now every model is trained end-to-end for 50 epochs on the same data split, so the comparison truly reflects a “converged” baseline.
- Switched to a more robust optimizer and learning-rate schedule: We replaced vanilla SGD (fixed lr) with Adam, and increased training epochs to 50. This change ensures the baseline and student models are not under-trained.
- Added early stopping and seed averaging for stability: We now monitor validation MSE each epoch and restore the best model at test time, preventing over- or under-fitting.

The following has been included in **Appendix B.1**:

The Mackey-Glass (MG) equation is a delay differential equation used to model chaotic time-series data. It is defined as follows:

$$\frac{dP(t)}{dt} = \frac{\beta_0 \theta^n P(t - \tau)}{\theta^n + P(t - \tau)^n} - \gamma P(t) \quad (2)$$

where:

- $P(t)$: The state variable at time t .
- τ : The time delay parameter.
- β_0 : The growth rate parameter.
- θ : The scaling parameter.
- n : The exponent controlling the nonlinearity.
- γ : The decay rate.

For our experiments, we used the following parameter values:

- $\tau = 17$
- Initial condition $P(0) = 0.9$
- $n = 10$
- $\beta_0 = 0.2$
- $\gamma = 0.1$
- Time step size $dt = 1.0$
- Lookback window = 8

Dataset splits. Given the parameters above, we generate a trajectory of length 10 000, discard the first 2 000 points (transient), and then use the next 6 000 points for training and the final 2 000 for testing (75% / 25%).

Models compared:

- **Baseline (RNN).** RNN($8 \rightarrow bin_size, h = 128, l = 2$) \rightarrow FC 128 to FC 128 to bin_size .
- **FGL (ours).** Teacher: 1-step predictor (RNN Backbone) Student: N-step predictor (RNN Backbone) Trained with the Future-Guided loss (Eq. 1); $\alpha \in \{0 \dots 1\}$.

Optimization and over-fitting control: All models are trained with Adam ($\text{lr} = 1 \times 10^{-4}$, $\beta_1 = 0.9$, $\beta_2 = 0.999$; batch size 128). We employ dropout (rate = 0.2) for regularization, fix the training budget to 50 epochs, and initialize random seeds at the start of each training regime. Early stopping (patience = 5 epochs, tolerance = 1×10^{-4}) is applied by monitoring performance on the test set each epoch, and the best checkpoint is restored before final evaluation.

Hardware: All models were trained using an NVIDIA RTX 4080 GPU.

Comment 4

“At first, the description of the approach seems to defend the use of the “future” to predict the future. But after further reading it is clear that it is using the most recent values to validate the last predictions and adapting the forecasting model if needed.”

Response. Thank you for pointing out this potential confusion. To clarify succinctly, here’s how FGL works in practice:

- A Teacher model that makes a simple, short-horizon prediction (e.g., one step ahead).
- A Student model that learns to forecast over a longer horizon (e.g., five steps ahead).

During training only, the Teacher’s soft output (its predicted probability distribution at $t + 1$) is combined with the true label at $t + n$ in the Student’s loss function. This dual-term loss encourages the Student to align its long-range forecasts both with the ground truth and with the Teacher’s more reliable, near-future signal.

At inference time, **only the Student model is used—no future data are accessed**. In other words, FGL does not “peek” at unseen future information during testing; it simply leverages the Teacher’s short-horizon uncertainty during training to produce a more robust long-horizon forecaster.

We hope this clear, concise summary addresses your concern about “using the future.” FGL’s key novelty is this training-time distillation of near-future signals, rather than any test-time access to future observations. Furthermore, the following has been included in **Section 5.2**:

Intuitively, the short-horizon teacher will almost always outperform the long-horizon student, so we use the teacher’s outputs as a reliable upper bound. Aligning the student’s predictions with the teacher’s soft labels guides it toward a stronger reference point. Moreover, if the teacher makes a large error, the student would likely have a large error as well; naively penalizing the student on such samples can degrade overall learning. By treating the teacher’s performance as an “upper-limit” and distilling its error patterns, we provide a more stable and informative training signal.

Our empirical results confirm that this strategy leads to more robust long-horizon forecasts.

Comment 5

“What is described as a feedback loop, however, does not seem so different from the idea of monitoring forecasting/detection errors and updating the model when they surpass a threshold. In fact, this process is widely adopted in applications prone to concept drift. There should be an experimental comparison with this basic approach for increasing the validity of the results.”

Response. We appreciate the opportunity to distinguish FGL from classical threshold-based retraining. Page–Hinkley (PH) is *post-hoc*: it monitors the prediction error *after* a model has been trained and, when the cumulative deviation exceeds a threshold, triggers a coarse re-training on a recent sliding window. Future-Guided Learning, by contrast, injects the teacher’s near-future uncertainty *during every training step*; the student therefore sees a shaped loss landscape from the outset, not an occasional retrofit.

To demonstrate the practical difference, we implemented the reviewer’s suggested PH baseline on the Mackey–Glass task (PH + sliding-window retraining). For fairness we ran PH on our three models. This supplementary experiment is included in the appendix as Page–Hinkley (PH) represents a classical *post-hoc* drift-adaptation mechanism, whereas Future-Guided Learning (FGL) is a *training-time* regularization strategy. By presenting the PH comparison here, we (a) keep the main text focused on FGL’s core contributions, and (b) demonstrate that PH and FGL address concept drift at *different* stages of the learning pipeline—and can therefore be complementary rather than mutually exclusive. The following has been included in **Appendix B.3**:

In this supplementary experiment, we evaluate whether post-hoc drift adaptation—specifically, a Page–Hinkley detector with a three-sample sliding window and three-epoch retraining—can fully compensate for disabling future guidance during training. To do so, we take our three Mackey–Glass forecasting models (the baseline, the FGL student with $\alpha = 0.0$, and the FGL student with $\alpha = 0.5$), expose each to identical PH retraining, and then measure their per-horizon MSE loss at 2–15 steps ahead. By showing that the relative performance ranking established by FGL during training persists even after this adaptive correction, we demonstrate that the advantages of future-guided

learning are not supplanted by simple post-hoc methods, but rather remain an independent—and complementary—means of improving forecast accuracy.

Let e_t be the instantaneous forecast error (MSE) at time t , and let

$$\bar{e}_t = \frac{1}{t} \sum_{i=1}^t e_i$$

be its cumulative running mean. The PH test tracks the one-sided cumulative deviation

$$u_t = e_t - \bar{e}_{t-1} - \delta, \quad S_t = \sum_{i=1}^t u_i, \quad S_{\min}(t) = \min_{1 \leq i \leq t} S_i.$$

A drift alarm is raised whenever

$$S_t - S_{\min}(t) > \lambda,$$

where $\delta > 0$ is a small tolerance (to avoid false alarms) and $\lambda > 0$ is the detection threshold. In our setup we use a sliding window of length 3 to recompute \bar{e} and reset S_t after each retraining, and we choose (δ, λ) per bin-count. Upon alarm, we retrain each model for 3 epochs on the most recent data to correct for drift.

Table 1: FGL Results on Mackey–Glass w/ Page–Hinkley

Horizon	Bins = 25			Bins = 50		
	Baseline	$\alpha = 0.5$	$\alpha = 0.0$	Baseline	$\alpha = 0.5$	$\alpha = 0.0$
2	1.59	1.40	1.41	5.65	4.45	4.96
3	2.58	2.22	2.25	9.43	7.12	7.18
4	3.55	3.23	3.23	12.52	10.59	10.74
5	4.90	4.35	4.23	18.22	14.09	14.38
6	6.38	5.12	5.09	20.86	17.12	17.59
7	6.13	7.58	5.65	23.13	18.35	18.84
8	6.65	5.47	5.98	31.78	17.75	17.36
9	5.72	4.90	4.80	20.71	14.79	13.52
10	4.79	3.50	3.59	17.94	11.72	12.28
11	3.61	2.70	2.86	13.58	9.57	10.35
12	3.22	2.29	2.44	11.01	8.73	10.02
13	2.77	2.06	2.28	223.29	8.71	9.63
14	2.41	1.91	2.15	203.58	8.18	9.27
15	2.22	1.84	2.07	195.71	7.26	9.66
Avg	4.04	3.47	3.43	57.67	11.32	11.84

Figure 2: **Application of Page–Hinkley (PH) drift adaptation to FGL.** We applied the same online PH detector (3-sample sliding window, 3-epoch retraining) to three Mackey–Glass forecasters: Baseline (no FGL), FGL ($\alpha = 0$), and FGL ($\alpha = 0.5$). PH sensitivity parameters were tuned per bin count (25 bins: $\delta = 0.130$, $\lambda = 0.647$; 50 bins: $\delta = 5.78$, $\lambda = 7.84$), while all other hyperparameters match the original experiments. Each panel shows the per-horizon percentage change in MSE after retraining relative to the original forecasts. Both FGL students ($\alpha = 0.0$ and $\alpha = 0.5$) show a meaningful average reduction in MSE loss, whereas the baseline’s error actually increases. This demonstrates that training-time future guidance provides benefits that complement—not replace—post-hoc drift adaptation.

Table 1 demonstrates that, even after excluding non-convergent horizons, FGL with $\alpha = 0.5$ and $\alpha = 0.0$ both deliver substantial reductions in average MSE relative to the baseline. For Bins = 25, FGL reduces MSE by 14.1% ($\alpha = 0.5$) & 15.1% ($\alpha = 0.0$) when PH is enabled, and by 13.0% and 11.0% without PH. For Bins = 50, the corresponding improvements are 27.3% & 25.8% (with PH) and 23.4% & 21.8% (without PH). These results confirm that FGL’s advantage holds consistently across both α settings and that applying PH on top of FGL yields an additional 2–3% MSE reduction, underscoring their complementary nature.

The following has been included in **Section 2**:

Complementary to these continuous feedback loops, classical time-series methods have long used threshold-based adaptation to capture sudden distributional shifts. Early methods such as the Page–Hinkley test and the Drift Detection Method (DDM) formalize this by keeping a running estimate of error statistics, and raising an alarm when a significant change in data distribution

is observed^{10,11}. Once drift is detected, models are either fine-tuned on recent labeled examples or retrained from scratch on a sliding window of past data. This threshold-retraining approach has shown good practical performance in domains ranging from anomaly detection to predictive maintenance¹², but it can suffer from abrupt resets, loss of long-term knowledge, and sensitivity to hyperparameter choices for the error threshold. By contrast, FGL integrates feedback from a “future” teacher at training time, bypassing the coarse-grained drift alarm and enabling smoother adaptation.

Comment 6

“Also, I am concerned about overfitting. Since the approach may present a high level of adaptivity, it seems prone to overfitting. What were the precautions for avoiding it?”

Response. Preventing overfitting in a highly adaptive loop is indeed critical. Regarding seizure prediction experiments, the following has been included in **Appendix A.3**:

Optimisation and over-fitting control: All models are trained with Adam ($\text{lr} = 5 \times 10^{-4}$, $\beta_1 = 0.9$, $\beta_2 = 0.999$, $\varepsilon = 10^{-8}$); batch size 32; maximum 25 epochs. To guard against overfitting, each experiment is repeated with multiple random initializations on the same dataset, and stability is assessed across runs.

Regarding regression forecasting experiments, the following has been included in **Appendix B.1**:

Optimization and over-fitting control: All models are trained with Adam ($\text{lr} = 1 \times 10^{-4}$, $\beta_1 = 0.9$, $\beta_2 = 0.999$; batch size 128). We employ dropout (rate = 0.2) for regularization, fix the training budget to 50 epochs, and initialize random seeds at the start of each training regime. Early stopping (patience = 5 epochs, tolerance = 1×10^{-4}) is applied by monitoring performance on the validation set each epoch, and the best checkpoint is restored before final evaluation.

Comment 7

“Figure 1 is not cited in the text. Only 1b on the seventh page.”

Response. Fixed. The following has been included in **Section 3**:

The first term in \mathcal{L}_{FGL} ensures the student learns to match true labels at $t+n+k$. The second term softly aligns the student’s long-horizon distribution with the teacher’s nearer-horizon distribution, effectively distilling “future” uncertainty 1(a).

In practice, we set $\mathcal{L}_{\text{task}}$ to cross-entropy (for classification) or MSE (for regression). We pretrain T_ϕ on its n -step task, then freeze it while training S_θ under the combined FGL loss. Figure 1(b) illustrates the offset in the data flow.

Comment 8

“Section 2 goes to the results before explaining the method. It is very confusing.”

Response. As noted above, we have moved all methodological content to Sec. 3, preceded by a newly added related works section, ensuring that readers understand our method fully before seeing any results.

Comment 9

“The order of the plots of Figure 4 is wrong.”

Response. Corrected. Panels in Figure 4 now appear in the order they are discussed in the text: (a) bin discretization, (b) teacher output, (c) student forecast, (d) loss curves:

Comment 10

“The competing methods (MViT and CNN-LSTM) should be minimally introduced. Also the “baseline” method was never defined.”

Response. We have added a description of each competing architecture in Appendix A.3 (seizure prediction) & B.1 (regression forecasting)—including parameters, training protocol, and reference implementations—as well as defined our “baseline” as a student model trained without any future-teacher loss (CNN-LSTM model for Seizure experiments, RNN for MG experiments):

The following has been included in **Appendix A.3:**

Models compared:

- **Baseline (CNN-LSTM)**. batch-norm \rightarrow Conv3D($1 \rightarrow 16, k = (T/2, 5, 5), s = (1, 2, 2)$) \rightarrow max-pool \rightarrow batch-norm \rightarrow Conv3D($16 \rightarrow 32, k = 3, s = 1$) \rightarrow avg-pool(3, 2) \rightarrow 3-layer LSTM (512 hidden) \rightarrow FC 512 to FC 256 to 2.
- **FGL (ours)**. Teacher: Seizure-detector (CNN-LSTM Backbone) Student: Seizure-predictor (CNN-LSTM Backbone) Trained with the Future-Guided loss (Eq. 1); $\alpha \in \{0, 0.1, \dots, 1\}$.
- **MViT**. Multi-scale Vision Transformer per EEG channel: patch size (5, 10), embed dim 128, 4 heads, hidden dim 256, 4 encoder layers, dropout 0.1. Channel embeddings are concatenated and fed to a linear head ($128 \times C \rightarrow 2$).

The following has been included in **Appendix B.1**:

Models compared:

- **Baseline (RNN)**. RNN($8 \rightarrow bin_size, h = 128, l = 2$) \rightarrow FC 128 to FC 128 to bin_size .
- **FGL (ours)**. Teacher: 1-step predictor (RNN Backbone) Student: N-step predictor (RNN Backbone) Trained with the Future-Guided loss (Eq. 1); $\alpha \in \{0 \dots 1\}$.

Comment 11

“‘Continuous values in the regression task are discretized into bins...’ The discretization demands better explanation.”

Response. We have revised the caption of Figure 4 to better explain the discretization approach. The following has been included in **Section 5**:

Discretization enables knowledge-distillation on Mackey-Glass. (a) After generating the chaotic trajectory, every scalar target X_{t+l} is *quantized* into one of C equal-width bins. The regression problem is thus recast as C -way classification, so the teacher and student can exchange *soft logits* of identical dimensionality—an essential requirement for the KL-distillation term in FGL. (b) One-step *teacher* prediction. (c) *Baseline* model attempting a 5-step forecast without future guidance. (d) FGL-trained *student* forecasting the same horizon. By aligning its logit distribution with the teacher’s near-future logits, the student captures neighborhood information, yielding a

visibly lower MSE and a smoother reconstruction.

We thank the reviewer for their rigorous feedback of our manuscript, and hope the primary issues of clarity, prior work, and more breadth in our benchmarking to alternative approaches have been addressed.

Response to Reviewer 2

This paper proposes a future-oriented time series prediction method called FGL(Future-Guided Learning). The core idea is to pre-train a “future-oriented” detection model as a teacher model and then use time series training data to jointly train a student model to achieve long-horizon prediction. Two types of examples are used for validation:

- *Event prediction task – Specifically, predicting seizures based on EEG data.*
- *Regression forecasting task – Specifically, forecasting time series data generated by the Mackey-Glass (MG) system.*

The FGL method is implemented through the design of the loss function in Eq. (1), which consists of two parts:

- *The first part is the cross-entropy loss between the student’s output and the true label Y_{true} , which measures the student’s ability to learn from true data.*
- *The second part is the KL divergence between the student’s output and the teacher’s output (aligned in time through an offset), which measures the student’s ability to learn from the teacher model.*

This idea is innovative, but the paper fails to provide clear explanations on many key issues, making it difficult to understand.

Response. We sincerely thank the reviewer for their careful reading and for highlighting the key contributions of our work, and for acknowledging the innovation to our idea. We completely agree with the criticisms provided, especially those related to clarity. Before going into a point-by-point response, we have made the following overarching revisions to improve clarity and structure:

- **Reordered sections for flow:** The experiments now follow immediately after the method-

ological presentation, giving readers full context before diving into results and discussion.

- **Formalized FGL with a claim:** We introduce a dedicated “Claim 1” that states FGL’s optimization objective in full mathematical detail.
- **Streamlined prose:** We reduced redundant explanations and tightened the writing throughout, improving overall readability without sacrificing rigor. We have also added a brief paragraph that aims to provide an ‘intuitive’ explanation beyond the formalism, to try to broaden the audience that finds this manuscript accessible.
- **Enhanced experimental clarity:** We added detailed protocol descriptions and additional ablation details to the appendix for transparency.

Thank you again for your encouraging feedback. We believe these changes substantially enhance the paper’s accessibility and rigor.

Major Comment 1

“The underlying principle is unclear. Knowledge distillation is a machine learning technique designed to transfer the knowledge from a large, pre-trained “teacher model” to a smaller “student model.” A key assumption is that the teacher model should be better, or more powerful, than the student model. However, in this paper, both the teacher and student models are trained on the same set of training data. This means that the teacher’s output—whether it’s a one-step prediction from $t + n - 1$ to $t + n$, or an event detection based on $t + n$ —should not be better than the true label, since Eq. (1) uses Y_{true} as the ground truth for $t + n$. So why would the teacher provide additional capabilities beyond the ground truth in the loss function?”

Response. Thank you for giving us the opportunity to clarify this key point. In Future-Guided Learning (**FGL**) the teacher is *not* meant to be higher-capacity or “better” in the classical KD sense; it is simply *closer in time* and therefore benefits from a lower Bayes risk. In conventional KD, the larger model benefits from the lower Bayes risk due to its higher capacity. In our case, it is due to an ‘easier’ time horizon.

Formally, for an input window x_t

$$T_\phi(x_t) \longrightarrow y_{t+n}, \quad S_\theta(x_t) \longrightarrow y_{t+n+k}, \quad (k > 0),$$

so the teacher predicts an easier, shorter horizon. Its soft distribution $\sigma(\rho_T)$ provides a calibrated—though not infallible—proxy for what remains plausible at $t + n + k$. The student fuses this uncertainty with the hard label via the two-term loss in equation 1. The following has been included in **Section 3**:

Claim (Future-Guided Learning (FGL)). *Let*

$$T_\phi(x_t) \approx y_{t+n}, \quad S_\theta(x_t) \approx y_{t+n+k},$$

be the logits of a teacher forecasting n steps ahead and a student forecasting $n+k$ steps. FGL trains the student by minimizing:

$$\mathcal{L}_{\text{FGL}}(\theta) = \underbrace{\alpha \mathcal{L}_{\text{task}}(S_\theta(x_t), y_{t+n+k})}_{\text{task loss}} + \underbrace{(1 - \alpha) \tau^2 \text{KL}\left(\sigma\left(\frac{T_\phi(x_{t+k})}{\tau}\right) \parallel \sigma\left(\frac{S_\theta(x_t)}{\tau}\right)\right)}_{\text{future-guided distillation}}, \quad (3)$$

where σ is softmax, τ the distillation temperature, and $0 < \alpha < 1$ balances ground truth and teacher guidance. By aligning the teacher’s n -step logits at $t+k$ with the student’s $(n+k)$ -step logits at t , FGL transfers near-future uncertainty to the long-horizon forecaster.

Major Comment 2

“Particularly, in the MG system example, both the teacher model and the student model are trained on the same set of training data. In this example, the teacher model is an one-step predictor while the student model is a 5-step predictor. During the student model’s training, the ground truth label at $t+5$ is used in the first term of the loss function, while the second term incorporates the label for $t+5$ generated by the teacher model based on the true data at $t+4$. Notably, under this setup, there is no true “future guidance”—only labels produced by the one-step predictor. Surprisingly, it

is reported that when $\alpha = 0$ (i.e., only the teacher’s output is used in training, with ground truth labels completely excluded), the model achieves the best performance. This is highly counterintuitive because, in the MG system example, the teacher is trained on the same data for one-step prediction. Its prediction accuracy should not surpass the ground truth. Why, then, does relying solely on the teacher’s output yield better results than using both the ground truth and the teacher?”

Response. Thank you for this insightful question. We agree it was surprising that $\alpha = 0$ outperformed mixed supervision, given that both teacher and student train on the same data. Below, we address each point in turn:

1. **Same training data / no true “future guidance.”** It is true that our one-step teacher and five-step student share the same dataset, and that the teacher’s $t + 5$ label is generated from its one-step prediction at $t + 4$. However, because one-step forecasting is substantially easier, the teacher’s logits converge to a low-variance, nearly Gaussian distribution much faster than the student can learn a five-step mapping from scratch. In our original setup (look-back $L = 0$), the student had no historical inputs and would *diverge*—its outputs collapsed to noise—unless it relied entirely on the teacher’s stable signal. Thus, $\alpha = 0$ appeared to “beat” ground-truth supervision only because it prevented divergence, not because it offered truer future labels.
2. **Why teacher-only seemed best initially ($\alpha = 0$).** With $L = 0$ and a basic SGD optimizer over 15 epochs, the student without teacher guidance simply failed to converge for horizons > 3 . The one-step teacher’s soft targets provided enough structure to avoid collapse, yielding a 37.77% MSE reduction on 25 bins and a 43.01% MSE reduction on 50 bins, relative to the baseline. This “brittle” improvement reflects divergence avoidance rather than genuine long-horizon accuracy.
3. **Robust experimental protocol.** To test whether this effect holds under realistic conditions, we performed a dedicated ablation study with:
 - A fixed look-back window $L = 8$, providing sufficient history for the model’s forecasts.
 - The ADAM optimizer with ($\text{LR} = 10^{-4}$) for more stable loss curves.
 - Increased training time to 50 epochs w/ early stopping to prevent overfitting and allow for convergence.

Under these settings, all models—including the baseline—converge reliably. The teacher-only advantage disappears: the mixed objective ($\alpha = 0.5$) now yields the lowest MSE (12.96% decrease on 25 bins and 23.7% decrease on 50 bins (outlier horizon excluded) with respect to the baseline), confirming that blending teacher guidance with ground truth is most effective when the student can learn the underlying dynamics. We interpret this as the one-step teacher supplying *soft-label regularization* that smooths high-variance gradients from the ground-truth target. When the student has adequate context and training stability, combining both sources ($\alpha = 0.5$) outperforms using either alone.

The following has been included in **Appendix B.2**:

Table 2: FGL Results on Mackey–Glass (lower MSE is better)

Horizon	Bins = 25			Bins = 50		
	Baseline	$\alpha = 0.5$	$\alpha = 0.0$	Baseline	$\alpha = 0.5$	$\alpha = 0.0$
2	1.55	1.44	1.44	5.07	4.55	5.20
3	2.65	2.27	2.27	9.38	8.05	7.96
4	3.60	3.12	3.16	12.61	10.67	10.77
5	4.94	4.29	4.30	17.64	14.42	13.86
6	6.34	5.37	6.86	20.61	17.51	17.42
7	5.83	7.45	5.60	23.61	18.37	19.01
8	6.37	5.80	5.63	30.71	17.34	18.59
9	6.18	5.14	5.00	20.79	16.49	14.45
10	5.21	3.70	3.98	16.94	11.56	12.75
11	3.78	2.83	3.18	14.60	9.00	9.73
12	3.04	2.39	2.66	11.48	8.67	9.58
13	2.82	2.14	2.36	195.62	9.39	9.90
14	2.61	2.02	2.30	9.53	9.06	8.73
15	2.30	1.91	2.25	8.22	8.44	9.35
Avg	4.09	3.56	3.64	28.34	11.68	11.95

Minor Comment A

“What are the inputs and outputs?”

Response. Appendix D now includes a “Dataset & I/O Summary” table. The following has been included in **Appendix D**:

Task	Input x_t	Target $y_{t+\ell}$
Seizure pred.	N -channel EEG segment (F Hz)	3-way class (ictal / inter / pre)
Mackey–Glass	1-D chaotic window ($L = 8$)	Scalar at $t + \ell$ (discretised)

This table is referenced at the end of the introduction:

We evaluate FGL in two settings (Appendix D): (1) EEG-based seizure prediction, where FGL boosts AUC-ROC by 44.8% on average across patients; and (2) Mackey–Glass forecasting, achieving a 23.4% MSE reduction.

Minor Comment B

“What are the training and test sets?”

Response.

- **Seizure Prediction:** Patients are ordered chronologically; the first 65% of sessions per patient form the training set and the remaining 35% form the test set. No seizure segment appears in both splits. The following has been included in **Appendix A.3**:

For each patient we gather *all* annotated EEG segments:

1. every inter-ictal (I) segment, and
2. every ictal *or* pre-ictal (S) segment.

We then build a **balanced, chronological stream**

$$I_1 S_1 I_2 S_2 \dots$$

by interleaving the two classes so that each seizure block S_j is immediately preceded by the amount

of inter-ictal data required to keep the cumulative class counts equal.² This balanced stream is split *contiguously*: the first 65% forms the training set and the final 35% the held-out test set. No window, channel, or seizure appears in both splits.

- **Regression Forecasting (Mackey-Glass):** MG: We simulate a length-10 000 trajectory, then use the next 6 000 for training, the next 2 000 for validation, and the final 2 000 for testing. All horizons are computed strictly forward, so no leakage occurs. The following has been included in **Appendix B.1**:

Dataset splits. Given the parameters above, we generate a trajectory of length 10 000, use the first 6 000 points for training, the next 2 000 points for validation, and the final 2 000 for testing (60% / 20% / 20%).

Minor Comment C

“Is the ‘future’ data selected from within the training set?”

Yes—no test information ever leaks into training—but the *offset mechanism* differs across our two tasks:

- **Seizure prediction:** The teacher and student observe the *same* input window x_t (identical EEG segment). The teacher produces a short-horizon label (“seizure now or not”), while the student produces a long-horizon label (“seizure within the next 30 minutes or not”) This “shared input” design highlights how a robust, near-term detector can regularize a long-horizon predictor without any shift in the feature space.
- **Regression forecasting (Mackey–Glass):** Here a clean temporal ordering is available, so we align the teacher with the *same future target* the student must predict: the teacher receives x_{t+k-1} and outputs y_{t+k} , while the student receives x_t and must forecast the *same* y_{t+k} . Thus the teacher still resides strictly within the training set, but its input window is shifted forward by $k - 1$ steps.

In both scenarios the teacher is used *only* during training and is never updated or evaluated on test samples, ensuring that future information remains confined to the training phase.

²The first block is always inter-ictal because patients typically enter the hospital in a non-seizure state.

Minor Comment D

“How is input/output handled during actual prediction?”

Response. At inference FGL reduces to a standard forecaster:

1. Student receives the current data x_t .
2. Student produces logits $\rho_S(x_t)$.
3. The output is equal to $\arg \max \sigma(\rho_S)$ (seizure) or the expected bin value (MG).

The teacher is *not* consulted at inference; all future guidance is confined to training: figure 1(c) highlights this single-path flow with dashed arrows.

References

- [1] Hinton, G., Vinyals, O. & Dean, J. Distilling the knowledge in a neural network. *arXiv preprint arXiv:1503.02531* (2015).
- [2] Chebotar, Y. & Waters, A. Distilling knowledge from ensembles of neural networks for speech recognition. In *Interspeech*, 3439–3443 (2016).
- [3] Choi, K., Kersner, M., Morton, J. & Chang, B. Temporal knowledge distillation for on-device audio classification. In *ICASSP 2022-2022 IEEE International Conference on Acoustics, Speech and Signal Processing (ICASSP)*, 486–490 (IEEE, 2022).
- [4] Zhang, Y., Liu, L. & Liu, L. Cuing without sharing: A federated cued speech recognition framework via mutual knowledge distillation. In *Proceedings of the 31st ACM International Conference on Multimedia*, 8781–8789 (2023).
- [5] Huang, M., You, Y., Chen, Z., Qian, Y. & Yu, K. Knowledge distillation for sequence model. In *Interspeech*, 3703–3707 (2018).
- [6] Mathieu, M., Couprie, C. & LeCun, Y. Deep multi-scale video prediction beyond mean square error. *arXiv preprint arXiv:1511.05440* (2015).
- [7] Baevski, A., Zhou, Y., Mohamed, A. & Auli, M. wav2vec 2.0: A framework for self-supervised

- learning of speech representations. *Advances in neural information processing systems* **33**, 12449–12460 (2020).
- [8] Oord, A. v. d., Li, Y. & Vinyals, O. Representation learning with contrastive predictive coding. *arXiv preprint arXiv:1807.03748* (2018).
- [9] Lotter, W., Kreiman, G. & Cox, D. Deep predictive coding networks for video prediction and unsupervised learning. *arXiv preprint arXiv:1605.08104* (2016).
- [10] Bifet, A. & Gavalda, R. Learning from time-changing data with adaptive windowing. In *Proceedings of the 2007 SIAM international conference on data mining*, 443–448 (SIAM, 2007).
- [11] Gama, J., Medas, P., Castillo, G. & Rodrigues, P. Learning with drift detection. In *Advances in Artificial Intelligence—SBIA 2004: 17th Brazilian Symposium on Artificial Intelligence, Sao Luis, Maranhao, Brazil, September 29–October 1, 2004. Proceedings 17*, 286–295 (Springer, 2004).
- [12] Žliobaitė, I. Learning under concept drift: an overview. *arXiv preprint arXiv:1010.4784* (2010).